# Predictors of Perceived Posttraumatic Growth and Depreciation as Outcomes of Experienced Discrimination

**DOI:** 10.3390/bs16010041

**Published:** 2025-12-24

**Authors:** Adriel Boals, Elizabeth L. Griffith, Ruth L. King, Kiet Huynh, Jonathan Cajas

**Affiliations:** Department of Psychology, University of North Texas, Denton, TX 76203, USAruthking@my.unt.edu (R.L.K.); kiet.huynh@unt.edu (K.H.);

**Keywords:** discrimination, resilience, posttraumatic growth, trauma, core beliefs

## Abstract

Although perceived posttraumatic growth (PPTG) has been examined in a wide variety of potentially traumatic and/or adverse events, very few studies have examined PPTG in response to experienced discrimination. Further, there is a strong need to better understand the factors that contribute to psychological outcomes following discrimination. The purpose of the current study was to examine correlates and predictors of unique variance in both positive (PPTG) and negative [perceived posttraumatic depreciation (PPTD)] trauma-related outcomes in response to discrimination experienced. A sample of 323 undergraduates from the United States (ages 18–58) who have experienced discrimination completed an online survey. The majority of participants indicated experiencing racial discrimination (51%), followed by gender discrimination (17%), and religious discrimination (10%). The results revealed that event centrality, perceived injustice, core beliefs, and resilience were all significantly associated with both PPTG and PPTD. In multiple regression models, core beliefs and resilience were unique predictors of both PPTG and PPTD. Specifically, core beliefs positively predicted PPTG and PPTD, and resilience positively predicted PPTG and negatively predicted PPTD. We conclude that experienced discrimination can powerfully alter one’s core beliefs, leaving the individual both more vulnerable to psychological depreciation, but also grants an opportunity for potential growth. We believe these findings can help clinicians better understand how to help individuals struggling with experienced discrimination.

## 1. Introduction

Researchers have increasingly recognized that experiencing an adverse event can lead to potential long-term *positive* effects, commonly referred to as posttraumatic growth (PTG; [36]; [55]). PTG specifically refers to positive benefits in the domains of personal strength, appreciation of life, spirituality, new possibilities, and close relationships with others, as a result of trauma. We use the term Perceived Posttraumatic Growth (PPTG) based on empirical evidence that self-reports of PTG more accurately assess perceptions of growth, as opposed to veridical growth (sometimes also called ‘actual’ or ‘genuine’ PTG) ([9]; [33]; [43]). Correlates of PPTG include a range of positive outcomes, including positive affect, satisfaction with life, and positive well-being ([4]; [28]). PPTG has been reported in response to a wide variety of potentially traumatic or adverse events, including military combat ([26]), sexual assaults ([22]), working as a health care professional ([40]), interpersonal violence ([20]), cancer diagnoses ([38]), and natural disasters ([3]). A meta-analysis of PPTG studies found that 53% of participants reported moderate or higher levels of PPTG ([61]). Hence, PPTG is commonly reported following exposure to a wide variety of potentially traumatic or adverse events and is associated with a variety of positive outcomes.

One type of adverse event that has the potential to cause profound psychological change is experiencing discrimination ([31]). Discrimination refers to the behavioral expression of prejudice, involving unfair treatment based on group membership ([6]). Experienced discrimination can include a one-time, isolated event but most often includes an accumulation of events, including acts of implied prejudice, perpetual microaggressions, intergenerational trauma, and repeated exposures to explicit, overt discrimination ([60]). Research has consistently documented that experienced discrimination is associated with elevations in psychopathology, including Posttraumatic Stress Symptoms (PTSS), stress, and anxiety ([16]; [23]). Only a small number of studies have investigated posttraumatic growth (PTG) resulting from discrimination, with seven identified that directly address this topic. Among the seven studies on discrimination, two focused on religious discrimination, showing that perceived discrimination among Muslim American college students predicted PTG via posttraumatic stress symptoms ([56]), and that religious coping was positively associated with PTG among forcibly displaced Muslims ([2]). Two other studies examined gender identity-based discrimination: one found that transgender and gender diverse persons of color showed higher PTG than their White peers ([54]), and another found that bullying based on sexual identity predicted greater PTG among sexual minority individuals ([45]). Studies on racial discrimination demonstrated that racial identity attitudes and mindfulness predicted PTG among Black participants ([18]) and that PTG was associated with flourishing among Black college students ([25]). Finally, a study examining cumulative adverse events—including sexual harassment and discrimination—found that such experiences were associated with higher PTG among college students ([34]). Collectively, these findings highlight the limited but growing evidence for positive psychological growth following experiences of discrimination, underscoring the need for further research in this area.

A goal of culturally and contextually informed theorizing is to make certain that all possible relevant variables are given appropriate consideration ([59]). Given the potentially psychologically destructive nature of experiencing discrimination, we also consider potential *negative* impacts in the same domains that comprise PPTG. Negative impacts in these domains are collectively referred to as perceived posttraumatic depreciation (PPTD). PPTD specifically refers to, as a result of trauma, *deficits* in the same domains that comprise PPTG—new possibilities, personal strength, spirituality, appreciation of life, and relations with others. In our search, we could find only one study that examined PPTD in response to experienced discrimination. A study by [44] ([44]) examined PPTG and PPTD in a sample of people living with HIV. The results revealed high intraindividual variability in both PPTG and PPTD across a five-day period, and that resilience was positively associated with PPTG and negatively associated with PPTD. Considering the great potential for PPTD following experienced discrimination and the dearth of research on this topic, we see a pressing need for further examination of this construct.

A useful first step in understanding the relationships between discrimination and PPTG and PPTD is to examine potential predictors of PPTG and PPTD within samples that have experienced discrimination. Although many correlates may be relevant, four variables stand out based on both empirical and theoretical rationales. Empirically, prior studies have identified these variables as particularly robust correlates of PPTG and/or PTSS (which is highly associated with PPTD). The theoretical rationale is based on [55]’s ([55]) account of what must occur for PTG to take place. They state that PTG has the potential to occur when an individual experiences an event that causes significant psychological struggle that shatters their beliefs about the self and world, followed by a period of rebuilding their beliefs in a manner that reflects growth and positivity. The first such variable is event centrality, which is the extent to which an individual construes an adverse event as central to their identity ([8]). From a theoretical standpoint, event centrality is important because it reflects the extent to which the adverse event colors their views of the self and world. From an empirical standpoint, previous research has revealed that event centrality is one of the strongest correlates of PTSS ([11]; [24]), and predicts unique variance in PPTG and PTSS, even after controlling for a host of other variables ([13]; [46]).

The second variable is perceived injustice, which refers to post-event cognitions specific to appraisals of the severity of loss, blame, unfairness, and irreparability of loss ([52]). From a theoretical standpoint, perceived injustice is important because it in part reflects the extent to which the individual experienced a psychological struggle and challenge following the adverse event. From an empirical standpoint, perceived injustice has evidenced strong correlations with PTSS and predicts unique variance in PTSS ([57]). The third variable is core beliefs, which refers to the extent to which an adverse event caused major disruptions in an individual’s beliefs about the self and world ([15]). As previously noted, PTG occurs when an adverse event causes a shattering of core beliefs and the individual is able to, over time, rebuild their core beliefs in a manner that reflects positive change ([55], making core beliefs an important theoretical component of predicting PPTG. In terms of empirical evidence, core beliefs have been found to consistently correlate with PPTG ([15]).

The fourth variable is resilience, which refers to the ability to successfully adapt in the face of trauma (see [49] ([49]) for a detailed discussion). [55] ([55]) state that PPTG occurs following intense psychological struggle. Highly resilient individuals are less likely to experience such a struggle, while less resilient individuals are vulnerable to such struggles following adverse events, making resilience a potentially important theoretical predictor of PPTG. Indeed, previous research has found that resilience is a robust correlate of PTSS ([42]) and PPTG ([30]); germane to the current study, is a particularly important factor when coping with discrimination ([35]).

In the current study, we attempted to fill three gaps in the current literature on discrimination-related PPTG. First, we broadened our sample to all types of discrimination. Individuals are discriminated against based on a variety of attributes, and we endeavored to capture this variety in our sample. Second, no known study has examined discrimination-related PPTD. There is a plethora of studies that have examined relationships between experienced discrimination and more traditional negative trauma-related outcomes such as PTSS, depression, and anxiety ([39]; [41]). Although PPTD is highly correlated with PTSS and depression ([1]), PPTD is differentiated in that it examines negative impacts in the same key domains that comprise PPTG—appreciation of life, spirituality, personal strength, relating to others, and new possibilities. We argue that in a study examining PPTG, it is interesting to also include PPTD for purposes of direct comparison. Third, and perhaps most novel, we know very little about predictors of discrimination-related PPTG, and nothing about predictors of discrimination-related PPTD. In the current study, we include the four aforementioned variables (event centrality, perceived injustice, core beliefs, and resilience) to allow for examination of simple correlations between these four variables and PPTG and PPTD, and, perhaps more importantly, for the identification of variables that predict unique variance in discrimination-related PPTG and PPTD.

Our attempt to fill these three gaps in the literature led to two primary research aims. Our first aim is to examine simple correlations between the four potential predictors we identified and PPTG and PPTD. Our second aim is to examine which of the four potential predictors predict unique variance in PPTG and PPTD. Although PPTG and PPTD are our two primary outcome variables, we also included five additional secondary outcome variables—PTSD, depression, anxiety, stress, and insomnia. We included these secondary trauma-related outcome variables to widen our examination of the impact of experienced discrimination on mental health outcomes. PPTG and PPTD mostly assess the general well-being of the individual in the key areas of personal strength, appreciation of life, spirituality, new possibilities, and close relationships with others. However, experiencing discrimination, like other types of potentially traumatic events, can also have downstream impacts on other aspects of mental health. The inclusion of these five secondary outcomes will allow us to further document the mental health impact of experienced discrimination and, perhaps most importantly, examine potential predictors of these outcomes.

We had five pre-registered hypotheses. Our first hypothesis (H1) is that all four of our predictor variables (event centrality, perceived injustice, core beliefs, and resilience) and all five of our other trauma-related outcome variables (PTSS, stress, depression, anxiety, and insomnia) will significantly correlate with PPTD; we hypothesize all of these correlations will be positive, with the exception of the correlation with resilience, which we hypothesize will be negative. Our second hypothesis is that all nine aforementioned variables in H1 will similarly significantly correlate with PPTG; we hypothesize positive correlations with event centrality, core beliefs, resilience, and PTSS, and negative correlations with perceived injustice, insomnia, stress, depression, and anxiety. Although our pre-registered hypothesis was for a positive relationship between PPTG and resilience, recent work points out that the relationship between these two variables vary greatly across studies ([19]). Hence we would not be surprised if this specific hypothesis is not supported in our current study. Our third hypothesis is that, in a multiple regression model with our four predictor variables predicting PPTD, only perceived injustice and event centrality will be significant predictors. This prediction is based on the robustness of these correlations reported in previous studies ([46]; [57]). Our fourth hypothesis is that, in a similar regression model predicting PPTG, only event centrality and resilience will be significant predictors. Once again, this prediction is based on the robustness of correlations between these variables reported in previous studies. We decided to eliminate our fifth pre-registered hypothesis[note 1]. In regard to our sample type, we recruited a college student sample because (1) discrimination is highly prevalent on college campuses ([50]), (2) this sample is ideal when attempting to obtain a wide variety of discrimination types, and (3) research has found that, in trauma studies, results obtained from college student samples do not substantially differ from results obtained from other sample types ([10]).

## 2. Methods

### 2.1. Participants

A convenience sample of 323 undergraduates was recruited from the Psychology Subject Pool from a large Southwestern university in the United States in exchange for partial course credit in psychology courses. The average age was 20.73 years (SD = 4.54; age range 18–58 years). A total of 47 identified as cisgender men (14.6%), 240 as cisgender women (74.3%), nine as non-binary (2.8%), one as transgender man (0.3%), 0 as transgender woman (0.0%), one as Native American (0.3%), one as ‘other’ (0.3%), and 24 did not report data (7.4%). This larger percentage of women is typical for this subject pool. The racial/ethnic profile was 123 White (38.1%), 78 African American (24.1%), 94 Hispanic (29.1%), 37 Asian (11.5%), 11 multiracial (3.4%), two Hawaiian (0.6%), and three Native American (0.9%). The sum of these percentages exceeds 100 because participants were encouraged to check all that apply.

### 2.2. Measures

Discrimination Event. We created a checklist with which participants indicated which type of discrimination they experienced. The purpose of this checklist was to assist participants identify their experienced discrimination so they could complete the outcome measures in reference this specific experienced discrimination. The checklist was designed to mimic existing measures of exposure to potentially traumatic events, but for discrimination types. We specifically asked, ‘You indicated that you have, at some point in your life, experienced discrimination based on some aspect of your identity. Please indicate which aspect of your identity that you felt was the source of that discrimination’. Thus participants selected a discrimination event from their lifetime. The response options were race or ethnicity, religious or non-religious background, sex or gender identity, disability status, age, sexual orientation, socioeconomic status, pregnancy status, parental status, or ‘something else about you’.

Perceived Posttraumatic Growth (PPTG) and Perceived Posttraumatic Depreciation (PPTD). We assessed PPTG and PPTD using the Posttraumatic Growth Inventory-42 item version (PTGI-42; [5]). This measure includes 21 items from the Posttraumatic Growth Inventory ([55]) and 21 matched negatively worded items. An example of a matched pair is “I discovered that I’m stronger than I thought I was” and “I discovered that I’m weaker than I thought I was.” Respondents rate each item on a 6-point Likert scale ranging from 0 (I did not experience this change as a result of my crisis) to 5 (I experienced this change to a very great degree as a result of my crisis). Positive and negative items were separately summed to create PPTG and PPTD composite scores (possible range of 0–105). Higher scores indicate higher levels of PPTG and PPTD. The internal consistency in the current sample was α = 0.94 (PPTG) and α = 0.94 (PPTD).

PTSD Symptoms. We assessed PTSS using the 20-item Posttraumatic Symptoms Checklist for DSM-5 (PCL-5; [58]). Participants rate on a 5-point scale from 0 (*not at all*) to 4 (*extremely*) how much they have been bothered by each symptom of PTSD. Example items included ‘Being super alert or watchful or on guard’ and ‘Repeated, disturbing dreams of the stressful experience’. Higher scores indicate higher levels of PTSS. In the current study, the internal consistency was α = 0.94.

Event Centrality. We assessed event centrality using the 5-item Centrality of Event Scale (CES; [8]). Participants endorse the statements using a five-point Likert scale from 1 (*totally disagree*) to 5 (*totally agree*). Example items include ‘I feel this event has become a part of my identity’ and ‘This event permanently changed my life’. Higher scores indicate higher levels of event centrality. In the current study, the internal consistency was α = 0.90.

Core Beliefs. We assessed core beliefs using the 9-item Core Beliefs Inventory (CBI: [15]). Participants endorse the statements using a six-point Likert scale ranging 0 (*not at all*) to 5 (*to a very great degree*). Example items include ‘Because of the event, I seriously examined my beliefs about the meaning of my life’ and ‘Because of the event, I seriously examined my beliefs about my own value or worth as a person’. Higher scores indicate higher levels of disruption of core beliefs. In the current study, the internal consistency was α = 0.92.

Resilience. We assessed trait resilience using the 25-item Connor-Davidson Resilience Scale (CDRISC; [17]). Participants endorse the statements using a 5-point rating scale ranging 0 (*not true at all*) to 4 (*true nearly all of the time*). Example items include ‘I am able to adapt to change’ and ‘Things happen for a reason’. Higher scores indicate higher levels of resilience. In the current study, the internal consistency was α = 0.95.

Perceived Injustice. We assessed perceived injustice using the Injustice Experiences Questionnaire (IEQ; [51]), which includes 12 items reflecting different thoughts regarding the sense of unfairness individuals might experience with respect to their adverse event. Participants endorse the frequency of such thoughts using a 5-point Likert-type scale ranging from 0 (*never*) to 4 (*all the time*). Example items include ‘It all seems so unfair’ and ‘I just want to have my life back’. Higher scores indicate higher levels of perceived injustice. In the current study, the internal consistency was α = 0.91.

Insomnia. We assessed insomnia symptoms using the 7-item Insomnia Severity Index (ISI; [7]). The ISI measures the nature, severity, and impact of insomnia in the past two weeks. Participants used a five-point Likert scale to rate each item from 0 (*no problem*) to 4 (*very severe problem*). Example items include ‘Difficulty falling asleep’ and ‘Problem waking up too early’. Higher scores indicate higher levels of insomnia symptoms. In the current study, the internal consistency was α = 0.86.

Depression, Anxiety, and Stress. We assessed depression, anxiety, and stress via the 21-item Depression, Anxiety, and Stress Scale (DASS-21; [29]). Participants responded on a scale ranging 0 (*did not apply to me at all*) to 3 (*applied to me very much or most of the time*). Example items include ‘I found it difficult to relax’ and ‘I felt I was close to panic’. Higher scores indicate higher levels of depression, anxiety, and stress. Internal reliabilities were α = 0.84, 0.83, and 0.82 for the three subscales, respectively.

### 2.3. Procedures

The study was an online survey. The inclusion criterion was that participants must have experienced discrimination based on some aspect of their identity. There was no exclusion criteria. After completing a consent notice, participants indicated which type of discrimination they experienced. They were then instructed to refer to this specific event when completing the next set of questionnaires. These questionnaires included the PTGI-42, PCL-5, CES, and IEQ. Participants were then asked to describe their discrimination event in an open text box. Since we wanted participants to complete the next set of questionnaires NOT in reference to their discrimination event, we followed with the explicit instructions of ‘just refer to how you are doing in general’. These questionnaires included the CDRISC, ISI, DASS-21, and demographics. This procedure was approved by the IRB of the University of North Texas. The study was pre-registered at: https://osf.io/8q4b9/?view_only=3794e68b53e04edf94bd795b2f6a0421 (accessed on 21 August 2025) (anonymized link).

## 3. Results

Prior to analyses, data was examined for normality and missingness. All data on all variables was within acceptable ranges of skewness (±2) and kurtosis (±7; [27]). Little’s MCAR test revealed that all data was missing completely at random (all *p*s > 0.05). For all analyses, listwise deletion was implemented for missing data.

Some types of discrimination were selected at much higher frequencies than others. The majority of the participants indicated that the type of discrimination they experienced regarded race (51%), followed by gender (17%), religion (10%), sexual orientation (8%), age (6%), disability (4%), socioeconomic status (2%), parental status (1%), and something else (1%). Since race and gender were the only discrimination types to have sample sizes greater than 50, these are the only two discrimination types that we examine individually in our subsequent analyses.

To test our first hypothesis that all nine of our selected variables would significantly correlate with PPTD, we conducted a series of correlational analyses. As can be seen in Table 1, our hypothesis was supported. All nine variables significantly correlated with PPTD in the expected directions. To test our second hypothesis, we conducted a similar series of correlational analyses but substituted PPTG for PPTD. As can be seen in Table 1, our hypothesis was partially supported. PPTG was significantly correlated with all four predictor variables (event centrality, core beliefs, perceived injustice, and resilience) but only significantly correlated with one of the five other trauma-related outcome variables (PTSS). Further, we hypothesized a negative correlation between PPTG and perceived injustice, but this correlation was positive.

To assess differences in correlations among variables within individual subsamples, we conducted a series of correlation analyses for the overall sample (Table 1) and for both the racial discrimination subsample (*N* = 155) and gender discrimination subsample (*N* = 51; see Appendix A). We next conducted Fisher’s r-to-z transformations to assess differences in the strengths of these correlations. Nuanced differences emerged between the two subsamples, although we do not draw conclusions from these differences given the small size of the gender discrimination subsample in comparison to the racial discrimination subsample. More information regarding these results is available via a reasonable request to the corresponding author.

To test our third hypothesis that perceived injustice and event centrality will predict unique variance in PPTD, we conducted a multiple regression model that included our four predictor variables and PPTD as the outcome. Post hoc power analyses revealed we had 0.99 power to detect a medium effect size (*f*^2^ = 0.15) and 0.80 power to detect an effect size of *f*^2^ = 0.05. We utilize *f*^2^ as a measure of global effect size because this metric relates the overall (i.e., *R*^2^) and relative (i.e., 1 − *R*^2^) contribution of predictors in explaining overall model variance ([48]); that is, *f*^2^ is calculated as [*R*^2^/(1 − *R*^2^)]. The overall model was significant, *F*(4, 302) = 19.26, *p* < 0.001, *R*^2^ = 0.20. However, the results did not support the hypothesis. As can be seen in Table 2A, perceived injustice and event centrality were not significant predictors in the model. Instead, core beliefs (positive association) and resilience (negative association) predicted unique variance in PPTD. To assess the unique variance explained by each predictor, we next calculated squared structure coefficients (*r*^2^_s_; see Table 2A) by first utilizing standardized regression coefficients to calculate predicted values of the relevant DV and second correlating each predictor with the predicted value; this approach allows us to examine the unique contribution of each individual predictor in explaining variance in our DV and also accounts for multicollinearity ([37]). All four predictors explained a significant proportion of the variance in PPTD; that is, although event centrality was not a significant predictor in the overall model, event centrality explained 53% of the variance explained in PPTD. In other words, although the model explains 20% of the variance in PPTD, event centrality explains 10.6% of the overall model variance. Perceived injustice also explained significant variance but was not a significant predictor. Furthermore, although a significant predictor in the model, resilience explained the least amount of variance in the model, which indicates a suppressor effect; that is, perceived injustice and core beliefs were non-significant predictors in the model but evidenced strong structure coefficients, which altogether implies that these constructs are relevant to our outcome but do not explain variance in the overall model due to suppression by other predictors.

We next conducted the same regression model to predict PPTD but only included the subsample of participants who selected racial discrimination as their discrimination type (*n* = 156)[note 2]. Post hoc power analyses revealed we had 0.98 power to detect a medium effect size (*f*^2^ = 0.15) and 0.80 power to detect an effect size of *f*^2^ = 0.08. The model was again significant, *F*(4, 151) = 9.83, *p* < 0.001, *R*^2^ = 0.21. As can be seen in Table 2A, core beliefs once again was a significant predictor, but resilience no longer significantly predicted PPTD. Squared structure coefficients revealed the same pattern as in the overall model such that event centrality and perceived injustice were not significant predictors in the model but explained large proportions of variance. We next conducted the same regression model but included only participants who selected gender discrimination as their discrimination type (*n* = 52). Post hoc power analyses revealed we only had 0.53 power to detect a medium effect size (*f*^2^ = 0.15), but we had 0.91 power to detect a large effect size of *f*^2^ = 0.35. The model was again significant, *F*(4, 47) = 7.10, *p* < 0.001, *R*^2^ = 0.37. Resilience was the only significant predictor (see Table 2A). Squared structure coefficients indicated that each predictor variable explained a significant proportion of variance.

To test our fourth hypothesis that event centrality and resilience will predict unique variance in PPTG, we conducted the same regression model to predict PPTD, but we substituted PPTG as the outcome variable. We had the same statistical power for these analyses that we had when testing H3. The overall model was significant, *F*(4, 302) = 27.83, *p* < 0.001, *R*^2^ = 0.27. The results mostly supported the hypothesis. As can be seen in Table 2B, event centrality and resilience were both significant predictors of PPTG. Unexpectedly, core beliefs was also a significant predictor in the model. Squared structure coefficients revealed that each predictor variable explained a significant proportion of variance.

We next conducted the same regression model to predict PPTG but only included the subsample of participants who selected racial discrimination as their discrimination type[note 3]. As can be seen in Table 2B, core beliefs and resilience once again were significant predictors, but event centrality was not. Squared structure coefficients indicated that each predictor variable explained a significant proportion of variance. We next conducted the same regression model but included only participants who selected gender discrimination as their discrimination type. As can be seen in Table 2B, we obtained the same pattern of results we observed for racial discrimination, in that core beliefs and resilience were significant predictors. Squared structure coefficients indicated that each predictor variable explained a significant proportion of variance.

## 4. Discussion

The present study examined perceived posttraumatic growth (PPTG) and perceived posttraumatic depreciation (PPTD) following experiences of discrimination. Consistent with prior research on adversity more broadly, we observed substantial reports of both growth and depreciation. Both PPTG and PPTD were correlated with all four of our theoretically relevant predictors (event centrality, core beliefs, perceived injustice, and resilience), aligning with models positing that both positive and negative psychological changes stem from cognitive, emotional, and identity-based processing of adverse events ([32]; [55]). Further, PTSS was the only trauma-related outcome variable that was significantly related to PPTG, whereas PPTD was significantly related to all five trauma-related outcome variables (PTSS, insomnia, depression, anxiety, and stress). Some of these results replicate previous findings, and some present novel findings. Our findings that PPTG was not significantly related to most of the trauma-related outcome variables is not uncommon ([28]). We would not consider the size or direction of any of the correlations surprising, but we consider all of these observed correlations to be noteworthy because they extend these correlations in response to the specific adverse event of experienced discrimination.

When predicting PPTD, we hypothesized that event centrality and perceived injustice would account for unique variance in a multiple regression model. Surprisingly, it was core beliefs and resilience that predicted unique variance in PPTD. This finding diverges from studies in more traditional trauma contexts where event centrality and perceived injustice often predict distress ([8]; [24]). One explanation is multicollinearity: event centrality and core belief disruption overlap conceptually and empirically, potentially obscuring unique variance attributable to event centrality. We suspect that event centrality was non-significant due to its multicollinearity with core beliefs. Indeed, event centrality and core belief disruption share much conceptual overlap. However, squared structure coefficients revealed that event centrality does indeed explain a significant proportion of unique variance in these models, indicating that other variables are accounting for the variance contributed by event centrality. In other words, although results of linear regressions indicate that event centrality is not noteworthy in these models, further calculations indicate otherwise.

Our finding that PPTG correlated significantly only with PTSS—but not with insomnia, depression, anxiety, or stress—mirrors meta-analytic evidence showing inconsistent or null associations between PTG and broader distress indicators ([28]). These results support the interpretation that PPTG often coexists with ongoing distress rather than replacing it. In contrast, PPTD demonstrated robust correlations with all distress variables, consistent with research identifying depreciation as a marker of more severe maladjustment ([53]). By documenting these contrasting profiles specifically within discrimination-related adversity, our study contributes to a growing literature suggesting that discrimination may simultaneously elicit identity threat and meaning-making processes that can generate both vulnerability and growth (e.g., [47]).

The finding with core beliefs gives credence to [32]’s ([32]) model that an adverse event shattering one’s views of the self and world is a core cause of PTSS, which is highly related to PPTD. The fact that resilience was significant in the multiple regression model was unexpected. The simple correlation between these two variables was significant but small. When we examined these predictors in only those who experienced racial or gender discrimination, the pattern of results was similar, but resilience did not significantly predict PPTD for racial discrimination, and core beliefs did not significantly predict PPTD for gender discrimination. Our data suggests that low resilience and greater shattering of beliefs may independently predict PPTD. Thus in concert with general findings on resilience, our data shows that a lack of resilience may be a critical reason why some people experience lasting PPTD following adversity.

For PPTG, our results largely aligned with previous findings: event centrality and resilience predicted unique variance, consistent with literature showing that growth often arises when an event becomes highly self-defining and individuals exhibit adaptive coping resources ([55]). However, core belief disruption also emerged as a significant predictor, reinforcing the model that growth frequently follows from a profound challenge to one’s worldview ([32]). The fact that both core beliefs and event centrality predicted unique variance, despite the fact that they have a great deal of conceptual overlap and are strongly correlated, is noteworthy. Core beliefs disruption is more specific to the extent to which the adverse event led to a shattering of views of the self and world, whereas event centrality is more specific to the extent to which the autobiographical memory of the adverse event serves as a reference point for everyday inferences, a turning point in the life story, and a core component of personal identity ([8]). Although some studies have found belief disruption to be more strongly related to distress than to growth ([14]), our results suggest that within the context of discrimination, shattered beliefs may simultaneously elevate vulnerability to PPTD and open pathways to PPTG. The fact that event centrality was non-significant in the racial and gender discrimination subsamples contrasts with broader trauma research, where centrality reliably predicts PTG ([24]). One possibility is that discrimination experiences—for many participants—were chronic or repeated rather than single, discrete events. For chronic stressors, identity-central events may be less tied to specific episodic memories and instead diffuse across multiple experiences, attenuating their predictive value in regression models. Structure coefficients again showed that event centrality still carried meaningful variance, reinforcing the need for future research to consider cumulative or chronic discrimination rather than single-event conceptualizations.

Taken together, our findings support a dual-process interpretation in which discrimination threatens core beliefs and self-integrity (driving PPTD), yet also catalyzes meaning reconstruction and identity reorganization (driving PPTG). The simultaneous importance of core belief disruption and resilience across models suggests that discrimination—similar to traditional traumas—can function as an inflection point in personal narratives, altering identity, worldviews, and coping trajectories ([33]). Our results extend this work by demonstrating that these processes occur not only after acute trauma but also in response to discrimination.

The current study contains a number of noteworthy limitations. First, we assessed self-reports of PPTG and PPTD, which can reflect a number of constructs other than genuine PTG, such as reappraisal coping, adherence to a cultural script, changes in narrative identity, and violation of post-recovery expectations ([9]; [12]; [33]). Second, our assessment of experienced discrimination was limited. We opted to simply ask what type of discrimination was experienced using a single item, as opposed to using an existing, more robust assessment of the experienced discrimination. Relatedly, our discrimination measure asked participants to choose discrimination towards one identity, which did not allow us to assess intersectional discrimination that involves discrimination toward multiple intersecting identities. Third, our discrimination events were disproportionately composed of the discrimination types of race and gender. The base frequencies of the other types of experienced discrimination were too low to obtain adequate statistical power to examine differences by discrimination type. Hence, our ability to generalize to discrimination experiences outside of racial and gender-based discrimination is limited. Fourth, our sample consisted of college students, which have restricted ranges of age, socio-economic status, and limited non-cisgender frequencies. Perhaps more importantly, discrimination experienced by college students may have limits on the extent to which they generalize to acts of experienced discrimination outside this sample. However, as we pointed out earlier, results from studies of the impact of trauma using college student samples generalizes well to other trauma samples ([10]). It should also be noted that this study was one of numerous studies available to the participant pool. Students who selected our study may have biases in that they wanted to share about their discrimination experiences, and thus may on average have more severe discrimination experiences. Lastly, although we conducted analyses separated by gender, our relatively small sample of men limited our statistical power for these analyses.

## 5. Conclusions

Like many adverse events, we found that discrimination can be a double-edged sword, capable of producing both psychological destruction and growth ([13]). It is vital that clinicians working with clients who have endured discrimination understand the impact of these adverse events and understand how they might promote PPTG as a response to discrimination ([21]) while minimizing subsequent PPTD. Resilience-building interventions that address discrimination-based experiences may need to incorporate identity-affirming and worldview-reconstruction components, given our findings that discrimination can simultaneously shatter core beliefs and activate meaning-making processes unique to identity-relevant adversity. In contrast, interventions for other traumas often focus more on event-specific recovery and coping skills, as they are typically anchored in discrete incidents rather than the chronic, identity-based stressors characteristic of discrimination. Our results identified core beliefs and resilience as two key factors in the link between experienced discrimination and growth and depreciation. The fact that these two variables predicted both PPTG and PPTD underscore their importance in the recovery process following trauma and/or adversity. Core beliefs disruption is common following trauma, and our findings demonstrate that core belief disruption as a result of experienced discrimination is no exception. As such, experienced discrimination can powerfully alter one’s core beliefs about the world and self. The result is the individual is both more vulnerable to psychological depreciation but also granted an opportunity for potential growth. As [53] ([53]) point out, PPTG is more likely to occur when core beliefs are reexamined. We encourage clinicians to focus on how experienced discrimination has altered the client’s core beliefs, with an emphasis on potential positive changes and fostering resilience.

## Figures and Tables

**Table 1 behavsci-16-00041-t001:** Correlations among all variables.

Variable	M(SD)	2	3	4	5	6	7	8	9	10	11
1. PPTG	45.16(22.34)	0.24 ***	0.32 ***	0.37 ***	0.34 ***	0.43 ***	0.28 ***	0.11 *^ns^*	0.05 *^ns^*	0.07 *^ns^*	0.07 *^ns^*
2. PPTD	23.85(19.44)	–	0.44 ***	0.34 ***	0.37 ***	0.42 ***	−0.12 *	0.21 ***	0.30 ***	0.31 ***	0.29 ***
3. PTSS	40.94(16.14)		–	0.66 ***	0.69 ***	0.64 ***	−0.17 **	0.36 ***	0.50 ***	0.50 ***	0.50 ***
4. Event Centrality	16.87(6.89)			–	0.72 ***	0.70 ***	−0.05 *^ns^*	0.20 ***	0.40 ***	0.37 ***	0.43 ***
5. Perceived Injustice	14.86(10.46)				–	0.71 ***	−0.12 *	0.26 ***	0.45 ***	0.43 ***	0.46 ***
6. Core Beliefs	16.93(11.01)					–	−0.03 *^ns^*	0.23 ***	0.43 ***	0.44 ***	0.46 ***
7. Resilience	63.49(18.15)						–	−0.25 ***	−0.35 ***	−0.32 ***	−0.36 ***
8. Insomnia	9.93(5.90)							–	0.39 ***	0.38 ***	0.39 ***
9. Depression	4.50(3.77)								–	0.86 ***	0.88 ***
10. Anxiety	5.06(3.76)									–	0.83 ***
11. Stress	5.78(4.41)										–

* *p* < 0.05, ** *p* < 0.01, *** *p* < 0.001, *ns* = not significant (*p* > 0.05). PPTG = Perceived Posttraumatic Growth. PPTD = Perceived Posttraumatic Depreciation. PTSS = PTSD Symptoms.

**Table 2 behavsci-16-00041-t002:** (**A**). Multiple Regression Model with Perceived Posttraumatic Depreciation as Outcome Variable. (**B**). Multiple Regression Model with Perceived Posttraumatic Growth as Outcome Variable.

(**A**)									
	**All Discrimination (*N* = 305)**			**Racial Discrimination (*N* = 155)**			**Gender Discrimination (*N* = 51)**		
Predictor	B(SE)	β	*r* ^2^ * _s_ *	B(SE)	β	*r* ^2^ * _s_ *	B(SE)	β	*r* ^2^ * _s_ *
Event Centrality	0.15(0.22)	0.05	0.53 ***	0.14(0.31)	0.05	0.54 ***	0.59(0.61)	0.22	0.54 ***
Perceived Injustice	0.19(0.15)	0.10	0.64 ***	0.19(0.21)	0.10	0.63 ***	0.24(0.29)	0.18	0.62 ***
Core Beliefs	0.55(0.14)	0.31 ***	0.81 ***	0.64(0.21)	0.34 **	0.95 *	0.22(0.30)	0.15	0.51 ***
Resilience	−0.12(0.06)	−0.11 *	0.19 ***	0.04(0.08)	0.04	0.03 *	−0.23(0.11)	−0.25 *	0.58 ***
(**B**)									
	**All Discrimination (*N* = 305)**			**Racial Discrimination (*N* = 155)**			**Gender Discrimination (*N* = 51)**		
Predictor	B(SE)	β	*r* ^2^ * _s_ *	B(SE)	β	*r* ^2^ * _s_ *	B(SE)	β	*r* ^2^ * _s_ *
Event Centrality	0.49(0.25)	0.15 *	0.26 ***	0.29(0.32)	0.09	0.28 ***	−0.56(0.72)	−0.16	0.52 ***
Perceived Injustice	0.21(0.17)	0.10	0.22 ***	0.06(0.23)	0.03	0.29 ***	0.67(0.34)	0.38	0.60 ***
Core Beliefs	0.50(0.15)	0.25 **	0.36 ***	0.85(0.22)	0.40 ***	0.60 ***	0.97(0.35)	0.52 **	0.73 ***
Resilience	0.36(0.06)	0.30 ***	0.56 ***	0.29(0.08)	0.24 ***	0.46 ***	0.31(0.13)	0.26 *	0.08 *

* *p* < 0.05, ** *p* < 0.01, *** *p* < 0.001. *r*^2^*_s_* = squared structure coefficient.

## Data Availability

This study was pre-registered at: https://osf.io/8q4b9/ (accessed on 21 August 2025). This site contains all materials and data used in this study.

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
