# Peer review of "Predictors of Perceived Posttraumatic Growth and Depreciation as Outcomes of Experienced Discrimination"

_behavsci, 2025, doi:10.3390/bs16010041_

Round 1

Reviewer 1 Report

Comments and Suggestions for Authors

This manuscript presents a timely and relevant analysis with a substantial sample of college students, offering meaningful insights into the impact of experienced discrimination on post-traumatic growth and depreciation. The topic is well aligned with the aims of Behavioral Science, and the use of multiple regression provides a clear analytic framework for understanding the relationships among the variables of interest. The paper is well organized and well structured, and the authors effectively communicate the significance of their findings. The results contribute to the growing body of research on student behavior and provide useful implications for future work in this area. Overall, the manuscript offers a valuable contribution and will be of interest to researchers and practitioners alike.

Description of the Sample: The sample size of over 300 students is adequate for multiple regression; however, more details about the sample characteristics would help the reader understand the population and evaluate generalizability. I think it would be a good idea to include more detail on the recruitment strategy, discuss sampling biases, and, if available, include a demographic breakdown (year of study, program of study). 

The manuscript states that the students had experienced discrimination. It is unclear to me whether the experiences of discrimination occurred on campus while a student or across the participants' lifetimes. A sentence regarding this would add clarity. 

Some terms could be defined more clearly for readers unfamiliar with the specific subfield.

  • We use the term Perceived Post-traumatic Growth (PPTG) based on empirical evidence that self-reports of PTG more accurately assess perceptions of growth, as opposed to veridical growth (Boals, 2023; Park& Boals, 2021; Jayawickreme et al., 2014).

The discussion section could better integrate the findings with existing literature. Specifically, you state 'Some of these results replicate previous findings, and some present novel findings'. Expand more by comparing and contrasting your results with previous studies in more depth. Explain unexpected or non-significant findings. Deepening this analysis will help readers understand the contribution of the study.

Details: You may be missing a number in this sentence: In our search, we could find only ___
study that examined PPTD in response to experienced discrimination.

Author Response

Reviewer Comment 1: Description of the Sample: The sample size of over 300 students is adequate for multiple regression; however, more details about the sample characteristics would help the reader understand the population and evaluate generalizability. I think it would be a good idea to include more detail on the recruitment strategy, discuss sampling biases, and, if available, include a demographic breakdown (year of study, program of study). 

Author Response 1: Thanks. In the Participants section we added that our sample was recruited from the Psychology Subject Pool, which is standard and understood by most/all psychological researchers. 

In the 2nd-to-last paragraph of the Discussion where we discuss the limitations, we added, 'It should also be noted that this study was one of numerous studies available to the participant pool. Students who selected our study may have biases in that they wanted to share about their discrimination experiences, and thus may on average have more severe discrimination experiences.'

Reviewer Comment 2: The manuscript states that the students had experienced discrimination. It is unclear to me whether the experiences of discrimination occurred on campus while a student or across the participants' lifetimes. A sentence regarding this would add clarity. 

Author Response 2: Good point. In the Measures section where we describe the Discrimination Event, we added, "Thus participants selected a discrimination event from their lifetime."

Reviewer Comment 3: Some terms could be defined more clearly for readers unfamiliar with the specific subfield.

Author Response 3: In the opening paragraph, we added, 'PTG specifically refers to positive benefits in the domains of personal strength, appreciation of life, spirituality, new possibilities, and relationships with close others, as a result of trauma. "

In the third paragraph we added, "PPTD specifically refers to deficits in the same domains that comprise PPTG, as a result of trauma (i.e., negative impact in the domains of new possibilities, personal strength, spirituality, appreciation of life, and relations with others). "

Reviewer Comment 4: We use the term Perceived Post-traumatic Growth (PPTG) based on empirical evidence that self-reports of PTG more accurately assess perceptions of growth, as opposed to veridical growth (Boals, 2023; Park& Boals, 2021; Jayawickreme et al., 2014).

Author Response 4: We assume you want us to define veridical. We added, "...as opposed to veridical growth (sometimes also called ‘actual’ or ‘genuine’ PTG)"

Reviewer Comment 5: The discussion section could better integrate the findings with existing literature. Specifically, you state 'Some of these results replicate previous findings, and some present novel findings'. Expand more by comparing and contrasting your results with previous studies in more depth. Explain unexpected or non-significant findings. Deepening this analysis will help readers understand the contribution of the study.

Author Response 5: Great point. We made substantial revisions throughout the Discussion section that we believe drastically improve the extent to which we compare and contrast with previous literature. All such additions are highlighted in yellow in the revised manuscript.

Reviewer Comment 6: Details: You may be missing a number in this sentence: In our search, we could find only ___
study that examined PPTD in response to experienced discrimination.

Author Response 6: Good catch. We added the word 'one'.

Reviewer 2 Report

Comments and Suggestions for Authors
  1. The article addresses an important topic that is rarely analyzed in empirical research: post-traumatic growth after experiencing discrimination.
  2. The abstract lacks information about the country in which the research was conducted and the age of the participants.
  3. The introduction presents issues related to post-traumatic growth and discrimination in a coherent and transparent manner, as well as an overview of the few studies to date on the relationship between these two constructs. It would be worthwhile to broaden the definition of discrimination—in its current form, it focuses more on examples of discrimination than on understanding the concept.
  4. The research objectives and hypotheses are correctly presented and well justified.
  5. The research tools used are described in a concise and comprehensive manner – the description of each tool includes the number of items, the response scale, sample items, and Cronbach's alpha reliability.
  6. The research procedure is presented in detail, but the exclusion criteria for the analysis are missing.
  7. Participants—the country in which the study was conducted should be added. The description of the study group does not clearly indicate the number of non-binary people (2.8% or 0.3%?). There is a large disproportion between the number of women and men (cisgender)—it is worth adding what this is due to. It is also necessary to justify that in this situation, the division by gender in statistical analyses is justified and methodologically correct.
  8. The results are presented clearly and consistently, and their accuracy is unquestionable—except for the issue of gender, as indicated in point 7 of the review.
  9. The discussion is interesting and cognitively valuable, although it could be enriched with broader references to the literature. It would also be worthwhile to point out the practical implications of the results obtained.
  10. The article points out the limitations of the analyses conducted.
  11. At least brief conclusions should be added

Author Response

Reviewer Comment 1: The abstract lacks information about the country in which the research was conducted and the age of the participants.

Author Reply 1: We added this info to the Abstract. 

Reviewer Comment 2: It would be worthwhile to broaden the definition of discrimination—in its current form, it focuses more on examples of discrimination than on understanding the concept.

Author Reply 2: In the 2nd paragraph of the Intro, we added a formal definition of discrimination. 

Reviewer Comments 4: ...the exclusion criteria for the analysis are missing.

Author Reply 4: In the Procedures section we added that there was no exclusion criteria. 

Reviewer Comments 5: Participants—the country in which the study was conducted should be added. The description of the study group does not clearly indicate the number of non-binary people (2.8% or 0.3%?). There is a large disproportion between the number of women and men (cisgender)—it is worth adding what this is due to. It is also necessary to justify that in this situation, the division by gender in statistical analyses is justified and methodologically correct.

Author Reply 5: We added the country in the Participants section.

Good catch on the non-binary category listed twice. The second instance was supposed to say Native American. We make this correction.

In the Participants section, we added: This larger percentage of women is typical for this subject pool.

The last sentence of the Discussion section now states: Lastly, although we conducted analyses separated by gender, our relatively small sample of men limited our statistical power for these analyses. 

Reviewer Comments 6: The discussion is interesting and cognitively valuable, although it could be enriched with broader references to the literature. It would also be worthwhile to point out the practical implications of the results obtained.

Author Reply 6: An excellent point. Reviewer 1 made a similar request. We include a detailed response to Reviewer 1. The short version is that we made substantial additions to the Discussion section that we are confident that satisfy this concern and greatly improves the manuscript.

Reviewer Comments 7: At least brief conclusions should be added

Author Reply 7: We added a Conclusions section at the end of the Discussion. 

Reviewer 3 Report

Comments and Suggestions for Authors

Thank you very much to the authors for this carefully considered and well-written contribution.

Regarding the abstract, it may be strengthened by concluding with a statement on the significance of the study or its broader implications.

Some in-text references are not in alphabetical order (e.g., lines 5–6). Please revise accordingly to maintain consistency with citation guidelines.

In the third paragraph, after discussing the negative impacts, the use of parentheses appears unnecessary; the ideas could be integrated more fluently into the sentence structure without them.

The section on predictors should be more clearly connected to the preceding literature. Presenting this literature earlier, before introducing the current study would enhance the coherence and flow of the review. As it stands, returning to the literature after describing the current study interrupts the narrative progression. Additionally, the hypotheses may be more appropriately placed within the methodology section.

I also have a comment regarding language use. The manuscript employs a “we” voice despite being a quantitative study, and there is no accompanying reflexivity section. Passive voice might be more appropriate for this type of research unless the authors intentionally wish to maintain an active “we” voice. If the latter is the case, it would be beneficial to include a brief reflection on their positionality, particularly regarding their relationship to the topic, experiences of discrimination (if relevant), and their role throughout the research process.

Several unexpected findings emerged (e.g., perceived injustice not predicting PPTD, core beliefs predicting both PPTG and PPTD, resilience predicting PPTD only in some subsamples). The discussion acknowledges these but could further elaborate on the theoretical implications. For example, what does it mean conceptually that core belief disruption predicts both growth and depreciation? How might this dual pathway align with more recent critiques of PTG (e.g., illusory growth, narrative reconstruction)? Strengthening these links would enhance the contribution of the discussion section.

The manuscript appropriately acknowledges the small size of the gender discrimination subsample. However, the discussion occasionally draws interpretive claims about differences across subsamples. Given the low power and instability of estimates, a stronger cautionary statement may be needed especially when reporting significance or nonsignificance of individual predictors in those models.

The manuscript concludes with implications for clinicians, but these could be developed further. For example, how might resilience-building interventions differ when addressing discrimination-based experiences versus other traumas? Expanding this section slightly would enhance the applied value of the study.

The discussion currently flows thematically, but not always in alignment with the stated research aims. Restructuring the discussion so that findings are presented directly in relation to Aim 1 and Aim 2 may help strengthen coherence and readability.

Author Response

Reviewer Comments 1: Regarding the abstract, it may be strengthened by concluding with a statement on the significance of the study or its broader implications.

Author Reply 1: Good point. Done.

Reviewer Comments 2: Some in-text references are not in alphabetical order (e.g., lines 5–6). Please revise accordingly to maintain consistency with citation guidelines.

Author Reply 2: Good catch. Done. 

Reviewer Comments 3: In the third paragraph, after discussing the negative impacts, the use of parentheses appears unnecessary; the ideas could be integrated more fluently into the sentence structure without them.

Author Reply 3: Done.

Reviewer Comments 4: The section on predictors should be more clearly connected to the preceding literature. Presenting this literature earlier, before introducing the current study would enhance the coherence and flow of the review. As it stands, returning to the literature after describing the current study interrupts the narrative progression. Additionally, the hypotheses may be more appropriately placed within the methodology section.

Author Reply 4: We understand this request and perhaps why a reviewer may make this comment. However, after trying to write the Intro in the manner requested, the flow became worse. I did not like it. I believe the way we did it originally is not wrong or incorrect. It comes down to a stylistic preference, and as author of the paper, I prefer it the way I wrote it. I am not trying to ruffle any feathers here, but I am hoping as author of the paper that I get some artistic license here. Thanks for hopefully understanding. 

Further, I believe the request to put hypotheses in the Methods is erroneous. According to the APA Manual, the Introduction should “describe relevant previous research and explain how it informs the current work,” and it should conclude by stating the study’s purpose and hypotheses (APA Publication Manual, 7th ed., Section 3.4).  

Reviewer Comments 5: I also have a comment regarding language use. The manuscript employs a “we” voice despite being a quantitative study, and there is no accompanying reflexivity section. Passive voice might be more appropriate for this type of research unless the authors intentionally wish to maintain an active “we” voice. If the latter is the case, it would be beneficial to include a brief reflection on their positionality, particularly regarding their relationship to the topic, experiences of discrimination (if relevant), and their role throughout the research process.

Author Reply 5: I believe this comment is also in error. According to the APA Manual, authors should “Use the active voice as much as possible” (APA, 2020, p. 118), especially when it is clear who performed the action. 

Reviewer Comments 6: Several unexpected findings emerged (e.g., perceived injustice not predicting PPTD, core beliefs predicting both PPTG and PPTD, resilience predicting PPTD only in some subsamples). The discussion acknowledges these but could further elaborate on the theoretical implications. For example, what does it mean conceptually that core belief disruption predicts both growth and depreciation? How might this dual pathway align with more recent critiques of PTG (e.g., illusory growth, narrative reconstruction)? Strengthening these links would enhance the contribution of the discussion section.

Author Reply 6: Excellent suggestion. Both Reviewer 1 and Reviewer 2 made similar comments. As I detailed in my response to Reviewer 1, we made substantial additions in the Discussion section that we believe more than adequately address this concern and strengthen the manuscript.

Reviewer Comments 7: The manuscript appropriately acknowledges the small size of the gender discrimination subsample. However, the discussion occasionally draws interpretive claims about differences across subsamples. Given the low power and instability of estimates, a stronger cautionary statement may be needed especially when reporting significance or nonsignificance of individual predictors in those models.

Author Reply 7: Done. In the Results section we state, "Nuanced differences emerged between the two subsamples, although we do not draw conclusions from these differences given the small size of the gender discrimination subsample in comparison to the racial discrimination subsample.". Further, at the end of the Discussion we added a sentence that explicitly acknowledges the low sample size of men. 

Reviewer Comments 8: The manuscript concludes with implications for clinicians, but these could be developed further. For example, how might resilience-building interventions differ when addressing discrimination-based experiences versus other traumas? Expanding this section slightly would enhance the applied value of the study.

Author Reply 8: Excellent suggestion. In the Conclusions section, we added, "Resilience-building interventions that address discrimination-based experiences may need to incorporate identity-affirming and worldview-reconstruction components, given your findings that discrimination can simultaneously shatter core beliefs and activate meaning-making processes unique to identity-relevant adversity. In contrast, interventions for other traumas often focus more on event-specific recovery and coping skills, as they are typically anchored in discrete incidents rather than the chronic, identity-based stressors characteristic of discrimination."

Reviewer Comments 9: The discussion currently flows thematically, but not always in alignment with the stated research aims. Restructuring the discussion so that findings are presented directly in relation to Aim 1 and Aim 2 may help strengthen coherence and readability.

Author Reply 9: Thanks for this comment. We made numerous revisions to the Discussion section that we believe improved the flow, coherence, and readability of the section. 

Round 2

Reviewer 3 Report

Comments and Suggestions for Authors

Thank you to the authors for the time and effort invested in revising the manuscript.

Regarding the organization of the literature review, I fully understand the authors’ wish to retain their preferred stylistic approach. However, my earlier suggestion was not solely about stylistic preference but about clarity and readability. As authors, we often become deeply immersed in our own writing, which can make it difficult to recognize where readers may struggle to follow the narrative. Peer review serves precisely this purpose—to offer an external perspective on coherence and accessibility. My concern was therefore related to flow and comprehensibility for future readers, not simply to stylistic choices. While it remains the authors’ decision not to restructure this section, I note that the current organization still makes it somewhat difficult to track topic transitions and maintain continuity.

A second point concerns Comment 5, which I believe remains only partially addressed. My original comment was conceptual rather than grammatical. The use of a personal “we” throughout a quantitative manuscript on discrimination implicitly signals researcher positionality. In areas such as discrimination, identity, and worldview-based impacts, reflexivity has become increasingly expected, even in quantitative research (e.g., White et al., 2021; Williams et al., 2021). The authors’ response focused on APA’s recommendations for active voice, which is accurate but does not address the core issue: APA style does not preclude reflexivity, nor does it eliminate the need for transparency around researcher position when studying identity-based phenomena. The revised manuscript does not include reflexive commentary nor a justification for why positionality is not relevant in this context. Accordingly, this response does not fully resolve the conceptual concern.

With these clarifications noted, the manuscript has been substantially improved through the revision process, and with minor additional adjustments, it would be suitable for publication.

Author Response

Reviewer Critique 1: Regarding the organization of the literature review, I fully understand the authors’ wish to retain their preferred stylistic approach. However, my earlier suggestion was not solely about stylistic preference but about clarity and readability. As authors, we often become deeply immersed in our own writing, which can make it difficult to recognize where readers may struggle to follow the narrative. Peer review serves precisely this purpose—to offer an external perspective on coherence and accessibility. My concern was therefore related to flow and comprehensibility for future readers, not simply to stylistic choices. While it remains the authors’ decision not to restructure this section, I note that the current organization still makes it somewhat difficult to track topic transitions and maintain continuity.

Author Response 1: I reorganized the Intro to conform to the request. I did not highlight all of the changes in the paper because it was mostly copying and pasting different sections around.

Reviewer Critique 2: A second point concerns Comment 5, which I believe remains only partially addressed. My original comment was conceptual rather than grammatical. The use of a personal “we” throughout a quantitative manuscript on discrimination implicitly signals researcher positionality. In areas such as discrimination, identity, and worldview-based impacts, reflexivity has become increasingly expected, even in quantitative research (e.g., White et al., 2021; Williams et al., 2021). The authors’ response focused on APA’s recommendations for active voice, which is accurate but does not address the core issue: APA style does not preclude reflexivity, nor does it eliminate the need for transparency around researcher position when studying identity-based phenomena. The revised manuscript does not include reflexive commentary nor a justification for why positionality is not relevant in this context. Accordingly, this response does not fully resolve the conceptual concern.

Author Response 2: I am a bit at a lost at what to do here. I understand what you are asking for and I do not disagree. However, I did a search for the word 'we' in the paper and reviewed all 84 such instances. In every instance, we are simply stating what we did. For example, 'We conducted a regression model....". I guess I could change this to the passive voice and say, 'A regression model was conducted....", but I fail to see how such edits would change anything about positionality. Importantly, I could find no such instance in which we use the word 'we' in which we are interpreting the results, making conclusions, or any other statement that could reflect positionality. So I am genuinely not sure what you want from us here.

I can include a reflexivity section if you want, but my preference is not to include one. My reason is that our study is quantitative, and reflexivity sections are typically used in qualitative studies. If you do want us to write one, I have never been asked to do so, so I could use some guidance given that we have great hetereogeneity of potential positionality among our five authors. Specifically, two of the authors are female, one is an ethnic minority, one is an ethnic and sexual identity minority, and I am a White male. Do I identify each author and state their potential positionality based on their demographics? Do I ask each author? Just let us know what you want from us.